# Instruments of Child-to-Parent Violence: Systematic Review and Meta-Analysis

**DOI:** 10.3390/healthcare11243192

**Published:** 2023-12-18

**Authors:** Luis Burgos-Benavides, M. Carmen Cano-Lozano, Andrés Ramírez, Francisco Javier Rodríguez-Díaz

**Affiliations:** 1Department of Psychology, University of Oviedo, 33003 Oviedo, Spain; andres.ramirez@ucacue.edu.ec (A.R.); gallego@uniovi.es (F.J.R.-D.); 2Department of Psychology, University of Jaén, 23071 Jaen, Spain; mccano@ujaen.es; 3Department Nursing Azogues Campus, Catholic University of Cuenca, Azogues 030102, Ecuador

**Keywords:** meta-analysis, child-to-parent violence, reliability, adolescents, youth, reliability generalization

## Abstract

This systematic review and reliability generalization meta-analysis synthesized psychometric literature on instrumentation assessing child-to-parent violence published through September 2023 across four databases. In the screening, we identified studies reporting Cronbach’s alpha internal consistency estimates for the child-to-parent violence scales. The eligible reliability coefficients ranged from 0.610 to 0.930, mostly exceeding the minimum threshold of 0.700. Random-effects models calculated pooled Cronbach’s alphas separately for global, father-specific, and mother-specific subscales. The results demonstrated cumulative values of 0.83 (global: standard error = 0.0129), 0.800 (fathers: standard error = 0.0203), and 0.81 (mothers: standard error = 0.0179), denoting largely adequate reliability. However, significant between-study heterogeneity was observed. While the mean alpha levels seem acceptable for most tools, substantial variability coupled with the possibility of some studies violating reliability assumptions indicates that a conservative interpretation is warranted. Ongoing scale refinement and additional psychometric evaluations will strengthen the rigor methodology in this developing research domain. However, these results should be interpreted with caution, as there is a high level of heterogeneity, and it is possible that some studies have not verified the assumptions underlying Cronbach’s alpha.

## 1. Introduction

Child-to-parent violence (CPV) is a global problem [1], similar to other types of violence. Over the past 15 years, it has been studied exponentially [2,3,4]. However, this problem is not new, as it was first identified more than 60 years ago as the “battered parent syndrome” by Harbin and Madden [5]. Despite this background, CPV remains the least studied and understood type of family violence compared to more well-known categories, such as child abuse, intimate partner violence, and gender violence [6,7]. More research is still needed to advance the knowledge base and improve public understanding of CPV as a distinct phenomenon. Although prevalent, it has neither permeated public consciousness nor received comparable empirical attention compared to other forms of violence occurring within families.

CPV is a complex phenomenon that has prompted discussion on several key aspects. The first relates to the variability in the terminology used to describe violence perpetrated by a child against a parent. Secondly, arriving at a precise definition has proven to be challenging, with at least 39 studies making original contributions to defining CPV [8]. The proposed definitions may adhere to strict or relaxed criteria. One widely cited definition by Cottrell [9] conceptualizes CPV as “any act by a child that is intended to cause physical, psychological, or economic damage to gain power and control over a parent”. A subsequent definition proposed by Pereira et al. [10] expands the conceptualization of the parental role to include non-biological caregivers while excluding acts arising from altered mental statuses, impaired consciousness, or withdrawal syndromes.

The assessment of CPV represents a third central issue. Ibabe [11] identified 11 instruments reporting psychometric properties regarding CPV evaluation, highlighting 3 of them as the most promising in clinical and research contexts: the Child-to-Parent Violence Questionnaire (CPV-Q) [12], the Adolescent Child-to-Parent Aggression Questionnaire (CPAQ) [13], and Abusive Behavior by Children-Indices (ABC-I) [14]. However, the Conflict Tactics Scale Children-Parent (CTS-CP) remains the predominant tool for gauging abusive child-to-parent behaviors and provides a historical foundation for CPV research [15,16]. Current studies continue to employ the CTS-CP, or adaptations thereof, to assess the dimensions of physical and psychological violence [17,18].

The psychometric assessment of child-to-parent violence (CPV) remains a salient challenge in furthering our understanding of this phenomenon. As theoretical foundations continue to develop [19], most explanatory models involve latent variables. Robust assessment tools are integral to address the high variability in epidemiological data pertaining to CPV prevalence and incidence and its associated factors. Definitional issues, assessment criteria (from technical to zero tolerance), and the psychometric limitations of existing instruments contribute to this variability. Evaluating pertinent interaction and mediating variables through psychometrically validated measures will enable more refined preventative and interventional approaches tailored to CPV [11,20].

This study aimed to identify child-to-parent violence assessment tools that demonstrate optimal internal consistency reliability and model fit based on psychometric syntheses. Cronbach’s alpha is an index of reliability derived from interitem correlations. The comparative fit index (CFI) gauges goodness-of-fit by contrasting the chi-square for the hypothesized model against that of a null baseline model, assuming that there is no relationship between variables. Additionally, the root mean square error of approximation (RMSEA) quantifies the divergence between the observed data and what would be expected under an ideal theoretical model. By meta-analytically synthesizing reliability metrics and fit indices across existing applications of each scale, this study will enable reliability generalization and facilitate evidence-based scale selection tailored to varying populations and disciplines. The identification of methodologically rigorous CPV measures constitutes a salient research need and key knowledge gap that these planned psychometric syntheses seek to address by quantifying reliability and model fit on a multidimensional basis.

The overarching objective of this study was to systematically identify investigations that have assessed child-to-parent violence (CPV) using psychometric instrumentation. The specific aims were threefold: The first aim was to identify the key features of publications focused on CPV assessment based on parameters including the year of publication, journal name and quartile ranking, instrument utilized, sample size, number of scale items, reported internal reliability (Cronbach’s alpha), and any fit indices relating to the underlying factor structure. Secondly, we calculated pooled estimates of the reliability and fit indices that have been documented for CPV measures across studies. Finally, we evaluated the between-study heterogeneity in the reported metrics to discern variability across different assessment tools. This research will facilitate a deeper understanding of the psychometric state of the field of CPV assessment while uncovering potential areas in need of additional examination to further strengthen the methodological foundations in this domain.

## 2. Materials and Methods

### 2.1. Eligibility Criteria

The inclusion criteria were as follows: (1) quantitative studies examining child-to-parent violence among adolescents or young people; (2) studies utilizing one or more psychometric scales to assess CPV constructs; (3) studies reporting Cronbach’s alpha values for the full scale, either the father or mother scale, or both; and (4) studies published in all languages.

Exclusion criteria were as follows: (1) book chapters, theses, conference proceedings, abstracts, systematic reviews, meta-analyses, and other gray literature not subject to peer review; (2) studies sampling exclusively parental populations; (3) qualitative studies or those not employing psychometric instrumentation; and (4) studies that did not report the global alpha.

### 2.2. Information Sources

Four scientific databases were used as sources of information, including Web of Science, Scopus, PsycInfo, and PsycArticles, with access to the library of the University of Oviedo and University of Almeria. Four databases were selected for their international scientific relevance and high-quality coverage of social and health science studies. The bibliographies of the articles included in the review were reviewed to reduce exclusion bias due to coverage.

### 2.3. Search Strategy

The construction of the search phrase was carried out using the terms that are frequently used as keywords in CPV articles and in other systematic reviews and meta-analyses (child-to-parent violence, child to parent violence, child-to-parent aggression, child-to-parent abuse, adolescent-to-parent violence, violent child-to-parent, adolescent violence toward parents, parent abuse, children violence toward parents, adolescent to-parent abuse, violence against parents, children violence against parents, adolescent violence against parents, parent abuse offense, child–parent violence, child/parent violence, child–parent aggression, youth-to-parent aggression, youth-to-parent violence, youth-to-parent abuse, youth aggression toward parents, youth violence toward parents, child-to-mother aggression, child-to-father aggression, teenage violence toward parents, adolescent-to-parent aggression, adolescent–parent abuse, adolescent aggression toward parents, adolescent violence toward parents, adolescent abuse toward parents, child-to-father violence, child-to-mother violence, child-initiated family violence, adolescent-initiated parent abuse, battered parent, juvenile domestic violence, adolescent family violence, youth violence in the home, teen violence toward mothers, parents abused by children, adolescent violence in the home, parent-directed aggression, violence children against mothers, aggression toward mothers, aggression toward fathers, mother abuse, abuse toward mothers, filio–parental violence, violence by children toward parents, violence by adolescents toward parents, parents abused by their children, abuse of parents by their adolescent, violence by children against parents, violence by child to parent, violence by adolescent to parent, aggression by child to parent, parents victimized by their children, parental abuse, and child-to-parent violences).

The search strategy included Boolean operators and truncation. The search fields were tailored to each database. For the Web of Science database, the “Topic” field was searched, encompassing terms appearing in article titles, abstracts, and keyword indexing. Within Scopus, the title, abstract, and author-specified keyword fields were selected to conduct searches. For both the APA PsycInfo and PsyArticles databases, a broad “all fields” approach was applied, with our specified terms being searched for simultaneously across titles, abstracts, and full texts. Customizing search fields per database in this manner allowed for a comprehensive scoping of the literature while accounting for variability in database structures and indexing methods (see Appendix A).

### 2.4. Selection Process

Searches across the four named databases were conducted on 20 September 2023, adhering to the guidelines outlined in the PRISMA statement [21]. Initial screening identified duplicate records, both algorithmically and through manual verification of records, with an over 90% content overlap. Secondly, records focusing on non-CPV subjects were excluded. Systematic reviews, theoretical papers, and non-systematic narrative reviews were deemed ineligible at this stage.

The remaining records underwent a full-text review by two independent researchers, judging each against the predetermined inclusion criteria. Disagreements regarding coding decisions prompted deliberation until consensus was achieved, with judgments categorized as included, excluded, or requiring further discussion. Coding for this phase was facilitated using Rayyan application web [22] and classified into duplicate, deleted, included, excluded, and possibly. The reasons underlying these decisions were logged within the system. Finally, blinding was disabled, allowing researchers to cross-check the results and continue with the study (Figure 1).

### 2.5. Data Collection Process

Core bibliographic information for all eligible records was systematically extracted from the selected databases. Specifically, the retrieved data included author names, publication titles, source journals, and abstracts. One author downloaded these details in RIS file format at the individual record level per database. The RIS files were then collated and uploaded to the Rayyan Bibliographic Manager. An independent researcher was invited as a blinded coding partner to enable dual screening functionalities. This structured extraction and file-processing protocol was followed sequentially across the four databases.

### 2.6. Data Items

Two complementary outcome categories were evaluated: (1) psychometric properties assessing model fit, reliability, and dimensional structure of CPV measurement instrumentation, and (2) methodological characteristics of the identified studies to enable evaluation of validity and generalizability.

Specific variables encompassed in model property analyses were (1) internal reliability, operationally defined as the degree to which a scale produces consistent results over repeated assessments, using Cronbach’s alpha coefficient as the metric; (2) comparative fit index (CFI) gauging the goodness-of-fit of hypothesized structural models; and (3) root mean square error of approximation (RMSEA) quantifying divergence between hypothesized and data-driven models.

Methodological descriptors comprised (1) publication name/author details, (2) source journal and quartile ranking, (3) instrument utilized for CPV assessment and associated item count, and (4) sample nature and size (N). Additional model fit indices, including McDonald’s omega, Tucker–Lewis index (TLI), and standardized root mean square residual (SRMR), were omitted from syntheses because of the paucity of available literature furnishing these statistics.

### 2.7. Study Risk of Bias Assessment

Two independent reviewers assessed the eligibility and methodological quality of the included studies by examining the internal reliability of child-to-parent violence (CPV) psychometric tools using Cronbach’s alpha. A priori alpha values exceeding 0.70 were considered acceptable per widely adopted standards. Additionally, the study appraisal incorporated sample representativeness, and the potential risk of bias stemmed from insufficient sample sizes. Agreement between reviewers was initially high (86.6%), with discrepancies resolved via deliberation and consensus.

Of the 23 identified studies reporting a global Cronbach’s alpha for the full assessment scale, 7 studies were excluded (citations) due to suboptimal alpha values or inadequate sampling. Regarding the 16 studies documenting reliability statistics for the father and mother subscales, 4 studies were excluded from each group (citations) due to similar deficiencies in alpha estimations or sample size criteria. The primary methodological shortcoming among studies furnishing global scale alphas was an insufficient sample size, while small sample sizes and unacceptable alpha values contributed equally to exclusion decisions for the scales of the father and mother.

### 2.8. Statistical Analysis

This study conducted reliability generalization and meta-analytic syntheses to quantify the internal consistency and model fit metrics for child-to-parent violence assessment instruments across the published literature. Cumulative reliability was estimated using Cronbach’s alpha values transformed via the Hakstian–Whalen approach to enable random-effects modeling, with separate analyses for the global, father, and mother subscales [23]. Eligible studies also furnished model fit data including comparative fit index (CFI), Tucker–Lewis index (TLI), root mean square error of approximation (RMSEA), and standardized root mean square residual (SRMR). Heterogeneity was examined using Cochran’s Q, I2, H2, and τ2 metrics [24,25]. To gauge result robustness, the trimming technique removed studies violating criteria around alpha magnitudes > 0.700 or minimum criterion of 10:1 cases-per-item ratio and inadequate cases-per-item ratios prior to calculating pooled estimates. The risk of publication bias was assessed using visual funnel plot asymmetry and Egger’s regression test. By meta-analytically quantifying reliability and validity indicators across applications, this research enables generalization regarding the capability of available tools to provide consistent measurements across varying samples relevant to child-to-parent violence. All analyses were performed using Jamovi version 2.4.8 and R statistical software version: 2023.09.1+494 (Metafor package).

## 3. Results

The results of the 31 identified studies are shown in Table 1. Regarding the sample size characteristics, the average number of participants per study was 1177 ± 1477 (median = 1100; range: 34–8115). In terms of the scale properties, the selected tools for assessing child-to-parent violence consisted of an average of 13.84 ± 10.72 items (median = 10; range: 6–62 items). The mean Cronbach’s alpha coefficient across all studies (N = 31) was 0.84 ± 0.060, denoting high average internal reliability (median = 0.800; range: 0.700–0.930). The majority of the studies (*n* = 18; 58.06%) reported that the global Cronbach’s alpha statistics were aggregated across all scales, while a set provided reliability estimates separately among fathers (*n* = 13; 41.94%) or mothers (*n* = 13; 41.94%).

The pooled Cronbach’s alpha reliability estimates and confidence intervals were calculated using a random-effects model with a restricted maximum likelihood. The cumulative analysis synthesized data from 31 eligible publications released between 2010 and 2023, which reported global scale Cronbach’s values. The results demonstrated an overall alpha coefficient of 0.833 (standard error = 0.0129; 95% CI Lower Bound = 0.808; CI Upper Bound = 0.858), indicating high internal reliability. However, there was substantial between-study heterogeneity according to the *I*^2^ (98.45%) and *Q* statistics (*Q =* 1104.438; *p* < 0.001). The complete results are presented in Table 2 and summarized in Figure 2.

A total of 18 eligible studies reported the global Cronbach’s alpha, but 23 Cronbach’s alpha measures were found, and 6 measures were excluded because they did not meet the criteria of an alpha > 0.70 or a sample size of 10:1 cases-per-item ratio. Finally, analyses were performed using 17 Cronbach’s alpha measures. Egger’s regression test indicated significant asymmetry, which is consistent with the potential publication bias (intercept = −2.338, *p* = 0.019). The highest internal reliability estimate was reported by Calvete et al. [15], with an alpha of 0.93. However, the two measures yielded alphas below the minimum acceptable threshold of 0.70 (Calvete et al. [29], *α =* 0.610 and *α =* 0.680; Martín, and Cortina [38] *α =* 0.690) An additional three measures were excluded to provide inadequate cases-per-item ratios, violating the minimum sample size requirements relative to the scale length with regard to the studies by Ghanizadeh and Jafari [32] (27 items; *n* = 74), Martin et al. [37] (9 items; *n* = 89), and Zuñeda et al. [41] (6 items; *n* = 34). A visual inspection of a funnel plot including the remaining 16 eligible studies also denoted asymmetry, echoing the results of Egger’s test (−3.261, *p* < 0.001). In addition, it was observed that the most used instruments were the Child-to-Parent Aggression Questionnaire, Conflict Tactics Scales for Young Psychologists Association, Abused Parent Questionnaire, Child-to-Parent Violence Questionnaire, Intra-family Violence Subscale, Abusive Behavior by Children-Indices, Adolescents’ parent-directed aggression, Child-to-Mother Violence Scale, Child-to-Parent Violence Risk (Risk Assessment), Parent Abuse Scale (Girl-mother version), and Violent Behavior Questionnaire.

A total of 13 eligible studies reported Cronbach’s alpha values for father-specific scales, but 16 Cronbach’s alpha measures were found, and 4 measures were excluded because they did not meet the criteria of an alpha of > 0.70 or a sample size of a 10:1 cases-per-item ratio. Finally, analyses were performed using 12 Cronbach’s alpha measures. Egger’s regression test showed no evidence of significant publication bias based on funnel plot asymmetry (intercept = −1.072, p = 0.284). However, two studies yielded internal consistency estimates below the acceptable threshold of 0.700 (Lyons et al. [51], α = 0.650; Navas-Martínez and Cano-Lozano [53], α = 0.660). An additional investigation by Kuay et al. [49] was removed to violate the minimum sample size standard relative to the number of scale items. A random-effects meta-analysis was conducted to synthesize father-specific alphas from the remaining 12 studies. The results demonstrated a cumulative alpha coefficient of 0.80 (standard error = 0.0203; 95% CI Lower Bound = 0.763; CI Upper Bound = 0.843), indicating adequate reliability, albeit with considerable between-study heterogeneity per the I2 (98.92%) and Q statistics (Q = 92.167; p < 0.001). The visual and statistical examinations showed significant funnel plot asymmetry, which was consistent with the potential publication bias (Egger’s regression test: −3.283, *p* < 0.001). The results are presented in Table 2 and Figure 3.

A total of 12 measures were analyzed on a mother-specific scale. Funnel plot symmetry was observed, with Egger’s regression test indicating a lack of significant publication bias (intercept = −0.430, p = 0.667). However, three studies yielded reliability coefficients below the minimum threshold of 0.700 (Lyons et al. [51], α = 0.650; Navas-Martínez and Cano-Lozano [53], α = 0.660; Kuay et al. [49], α = 0.670). The study by Kuay et al. [49] was also removed for providing insufficient cases relative to the number of scale items. A random-effects meta-analysis was subsequently performed on data from the remaining 12 studies that examined mothers. A cumulative alpha of 0.810 was demonstrated (standard error = 0.0179; 95% CI Lower Bound = 0.777; CI Upper Bound = 0.847), reflecting adequate internal consistency, coupled with significant between-study heterogeneity per the I2 (98.66%) and Q estimates (*Q =* 74.787; *p* < 0.001). However, a visualization of a funnel plot incorporating these 12 studies showed significant asymmetry, corroborated by Egger’s regression test (−3.023, *p* < 0.001), indicating potential publication bias. The complete alpha reliability results for the mother-specific subscales are presented in Table 2, Figure 4 and Figure 5.

Descriptive statistics were computed for the model fit indices extracted from the 21 eligible studies. All included investigations (100%) furnished the comparative fit index (CFI) and root mean square error of approximation (RMSEA) values required for the random-effects meta-analysis. The mean CFI across the studies was 0.959 (standard deviation = 0.021), indicating largely adequate model fit on average, and exceeding the recommended minimum thresholds. The average RMSEA was 0.037 (standard deviation = 0.022), indicating satisfactory fit per the conventionally accepted cut-off. The complete descriptive results of the model fit indices are presented in Table 3

A total of 20 eligible studies satisfying the minimum 10:1 cases-to-item criterion were incorporated in the meta-analyses examining model fit indices for child-to-parent violence assessment tools. One investigation was excluded because of an inadequate sample size relative to the scale length (Ghanizadeh and Jafari [32]; 27 items; *n* = 74). Random-effects models were employed to synthesize comparative fit index (CFI) values across the studies.

The results demonstrated a cumulative CFI estimate of 0.960 (standard error = 0.00489; 95% CI Lower Bound = 0.951; CI Upper Bound = 0.970), denoting an excellent model fit, on average. However, tests of heterogeneity showed a substantial dispersion of effects per the I2 (99.93%) and Q statistics (*Q =* 1036.847; *p* < 0.001). Risk of bias assessments revealed a significant Egger’s regression intercept (−4.728, *p* < 0.001) coupled with the subjective asymmetric visualization of a funnel plot incorporating the 20 CFI study results. The complete model fit and meta-analytic details are shown in Table 3, Figure 6 and Figure 7.

A random-effects meta-analysis was conducted to synthesize root mean square error of approximation (RMSEA) values across 20 eligible studies gauging model fit for child-to-parent violence tools. The results demonstrate a cumulative RMSEA estimate of 0.026 (standard error = 0.0076; 95% CI Lower Bound = 0.011; CI Upper Bound = 0.041). This mean value fell below the 0.06 threshold recommended for an adequate fit; however, there was substantial heterogeneity according to the Q-statistic (Q = 5.337; *p <* 0.001). A risk of bias assessment through a visualization of a funnel plot coupled with Egger’s regression (intercept = −3.274, *p <* 0.001) suggested the presence of asymmetry, which is indicative of possible publication bias. The complete results of the RMSEA meta-analysis are presented in Table 3, Figure 7 and Figure 8.

## 4. Discussion

The assessment of child-to-parent violence (CPV) constitutes an emerging area of priority research interest, given the imperative to establish prevalence and severity benchmarks as a basis for targeted intervention programming [11,19]. Reliable and valid instrumentation remains fundamental to elucidating the presence, patterns, predictors, and outcomes associated with CPV through empirical research [11,55]. Aligned with these priorities, the current systematic review and meta-analysis aims to inform this rapidly developing field by synthesizing the conceptual underpinnings, operating characteristics, and psychometric evidence base of measures utilized to quantify CPV experiences and behaviors. By evaluating instrument reliability and structural validity, this study sought to facilitate the identification of methodologically rigorous tools to aid the investigation, profiling, risk assessment, and evaluation support surrounding CPV.

The overarching objective was to systematically review the literature for studies utilizing psychometric instrumentation to assess child-to-parent violence, which reports internal reliability metrics amenable to reliability generalization synthesis. The screening procedures identified 31 eligible investigations, providing 40 Cronbach’s alpha coefficient estimates (including longitudinal applications). The most frequently employed measure was the Conflict Tactics Scale (CTS; *n* = 10, 32.26% of the included studies), followed by the Child-to-Parent Violence Questionnaire and Child-to-Parent Aggression Questionnaire (both *n* = 7, 22.58% each). By quantitatively synthesizing reliability evidence across the diverse tools applied within this body of literature, this review facilitates instrumental comparisons of the capability of prominent scales to provide consistent reliability measures.

The child-to-parent violence assessment tools demonstrated largely adequate internal reliability as per the conventional Cronbach’s alpha standards, exceeding 0.70 across various cultural contexts, including Spain, the Midwest, Iran, Mexico, Canada and the United Kingdom, and Chile. Furthermore, the meta-analytic review identified selected studies within the past three years, furnishing alpha estimates surpassing 0.90, denoting exceptional consistency, including, specifically, three Spanish investigations [15,30,45], one from the United Kingdom and Canada [49], and one from Iran [39]. Such cross-validation of predominant CPV measures using reliability generalization techniques substantiates their utility as psychometrically robust instruments for the assessment of CPV.

Regarding the overall evaluation, a value of 0.80 is considered to be a solid estimate of the Cronbach’s alpha coefficient. According to George and Mallery [56], thresholds are established for Cronbach’s alpha: values below 0.50 are considered unacceptable, values between 0.50 and 0.60 are considered poor, and values between 0.60 and 0.70 are questionable, and therefore, values above 0.70 are considered acceptable, values above 0.80 are considered good, and values above 0.90 are considered excellent. In our analysis, no studies reported unacceptable values for internal consistency, although four studies presented estimates that were considered questionable [29,38,51,53].

Moderation analyses and variables of sex and age were also explored. However, these results did not indicate any significant effects. It is relevant to highlight that our findings show high heterogeneity; therefore, they should be interpreted with caution. This high heterogeneity did not appear to be influenced by the examined moderating variables. Although the minimum recommended number of studies was reached in most subgroup analyses, we cannot rule out the possibility that, with a larger number of studies, these moderators could explain the observed variability to some extent [57].

Despite the generalized reliability, significant heterogeneity was observed among the measurement instruments used in the studies included in the review. Some instruments seemed more sensitive to certain types of CPV or particular contexts. This raises the importance of carefully considering the choice of instrument based on the research objectives or study population, and the heterogeneity between instruments highlights the need for further research. The factors contributing to these differences in reliability and how they can be mitigated should be explored further. Additionally, the adaptation of existing instruments to specific populations or cultural contexts should be considered.

### Limitations

This study leads us to highlight that we must be aware that the Cronbach’s alpha has some limitations. One is that the Cronbach’s alpha is influenced by the number of items in the scale, which means that the greater the number of items in the scale, the greater the coefficient. In other words, the inter-correlation between items may be small even though the alpha is large. In contrast, the McDonald’s omega coefficient is not affected by the number of items in the scale, making the measure more reliable than the Cronbach’s alpha. A common mistake is that the Cronbach’s alpha reports unidimensionality; that is, a high alpha value is assumed to indicate the unidimensionality of the set of items, although it is not prudent to link the concept of unidimensionality with the Cronbach’s alpha without knowing if the set of items measures a single construct.

Furthermore, this study did not analyze the cultural characteristics and prevalence [58], which would have allowed us to more clearly discriminate the quality of the psychometric instruments. In future research, we intend to quantify the prevalence by country, explore the role of cultural phenomena, and investigate variables related to the independent living skills and autonomy of children who do not commit VFP, as well as different protective and preventive factors [59] and the use of positive conflict resolution strategies [60,61]. Finally, in the meta-analysis, several studies were eliminated for not meeting the criteria of 10 participants per item [62,63].

The current meta-analysis did not synthesize the McDonald’s omega coefficients, given the limited reporting of this reliability estimate across the identified literature on child-to-parent violence. These findings suggest that the Cronbach’s alpha remains the predominant metric employed in practice when evaluating and comparing the internal consistency of assessment tools. Research efforts furnishing both alpha and omega statistics are warranted to enable a more robust appraisal of the measurement precision. Additionally, expanded reporting of model fit indices, including CFI, TLI, RMSEA, and SRMR, would strengthen the psychometric evidence base surrounding emergent CPV instruments. An ongoing limitation precluding more fine-grained reliability generalization surrounds the paucity of factor-specific values, as opposed to global scale estimates, particularly considering contemporary multidimensional CPV definitions encompassing psychological, physical, and economic abuse subconstructs. The omission of factor-level reliabilities likely excluded several otherwise eligible investigations from the current synthesis, and they should be addressed in subsequent efforts to produce a more granular understanding of the reliability patterns underlying existing multidimensional assessment tools.

Another limitation of this study was that a sensitivity analysis was not conducted for the studies included in the meta-analysis. This is because there is currently no established algorithm for determining sensitivity groupings; therefore, the results may have been influenced by bias in the estimation of reliability in the estimated model. Overcoming this limitation should be a priority line of research to ensure adequate discrimination between CPV assessment tools and the robustness of the results (Appendix A).

## 5. Conclusions

In conclusion, this systematic review and meta-analysis critically examined the reliability and structure of child-to-parent violence (CPV) assessment tools. The synthesis findings substantiate adequate psychometric properties among the available instrumentation, while underscoring meaningful contextual and sample variables applicable to tool selection scenarios. Sustained research efforts remain vital for advancing the comprehension of CPV dynamics and directing effective evidence-based interventions using rigorous measurement models. Promising future directions include a cross-cultural adaptation of existing CPV scales, expanded validation initiatives aligned with COSMIN guidelines, the adoption of complementary reliability metrics such as McDonald’s omega, the elucidation of associated risk and protective predictors, and an ongoing outcome evaluation of preventative initiatives and therapeutic protocols. These meta-analytic results have valuable implications for both research and practice by documenting the utility of psychometrically validated quantification approaches that enable profiling, early risk detection, and the monitoring of CPV experiences in at-risk youth.

Although selected measures have demonstrated exceptional one-time reliability patterns, stability conclusions are constrained without additional longitudinal replications. Among the current instruments, the Child-to-Parent Violence Questionnaire (CPV) and the Child-to-Parent Aggression Questionnaire (CPA) exhibit favorable consistency, with Cronbach’s alpha values predominantly exceeding 0.80 across varying applications. Despite its prolific use, the Conflict Tactics Scale (CTS) evidence of greater variability in internal reliability estimates ranging from 0.60 to 0.70 warrants consideration. Presently, the CPA and CPV measures represent the most psychometrically promising options, given extant reliability patterns coupled with extensive global implementation in CPV samples. However, further evaluation using clinical and forensic groups is critical to complement the existing community-based evidence. Additional tools showing initial yet unprecedented reliability patterns similarly require re-evaluation in diverse populations (for example, the Abused Parent Questionnaire and the Explanations about Adolescent-to-Parent Violence Questionnaire, albeit only in single applications, thus far prohibiting stability conclusions).

## Figures and Tables

**Figure 1 healthcare-11-03192-f001:**
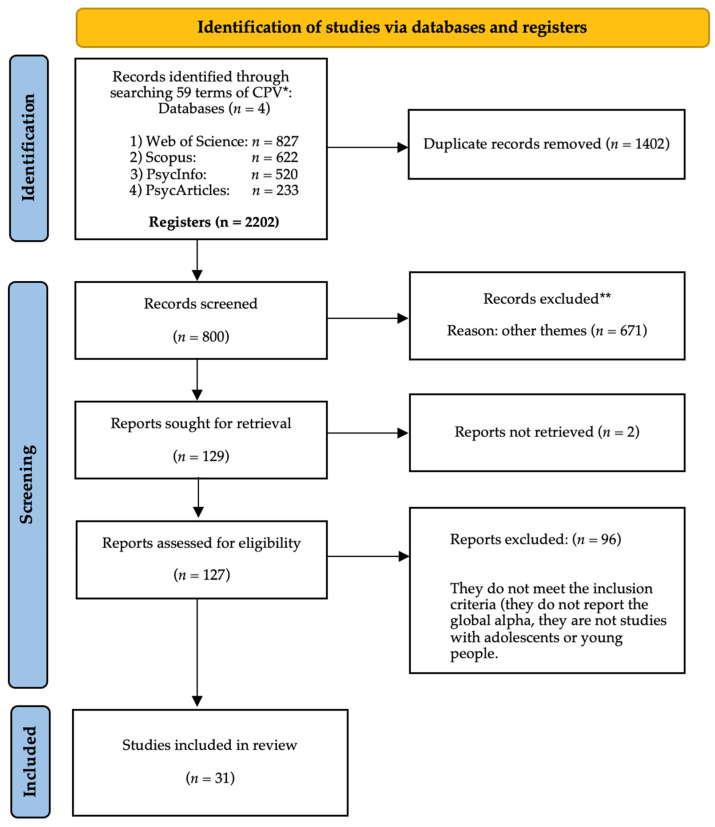
Flow diagram of the new systematic reviews. * Corresponds to the sum of all the records found in the information bases.; ** The records excluded without taking into account those duplicated by the criteria of a systematic review and other topics the study.

**Figure 2 healthcare-11-03192-f002:**
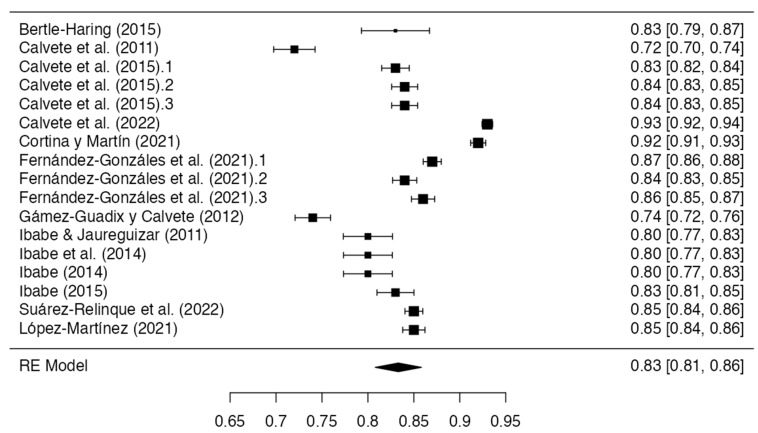
Forest plot of the reliability coefficients in the articles that reported Cronbach’s alpha globally (*n* = 17) [7,15,26,27,28,30,31,33,34,35,36,39,40].

**Figure 3 healthcare-11-03192-f003:**
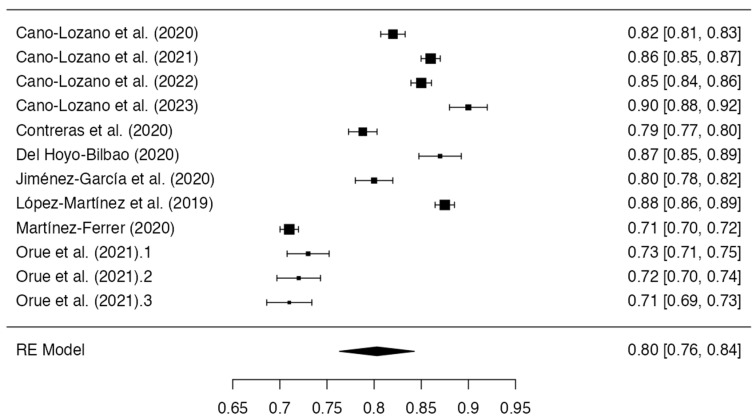
Forest plot of the reliability coefficients in the selected studies that presented the Cronbach’s alpha of fathers (*n* = 12) [42,43,44,45,46,47,48,50,52,54].

**Figure 4 healthcare-11-03192-f004:**
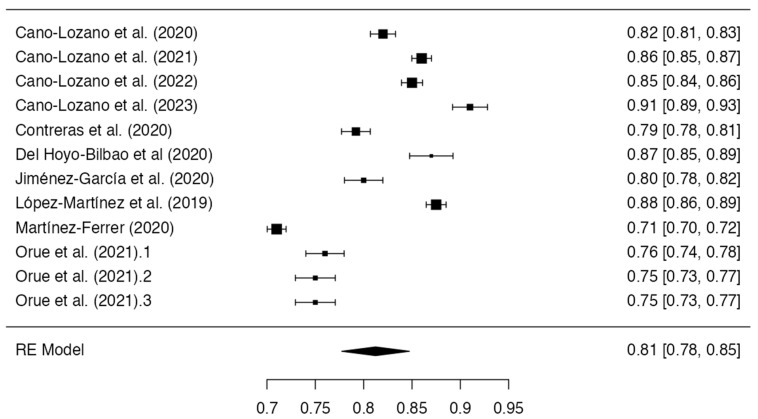
Forest plot of the reliability coefficients in the selected studies that presented Cronbach’s alpha of the mothers (*n* = 12) [42,43,44,45,46,47,48,50,52,54].

**Figure 5 healthcare-11-03192-f005:**
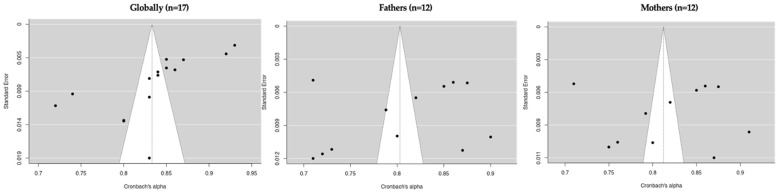
Funnel plot of all studies selected for meta-analysis of Cronbach’s alpha coefficient.

**Figure 6 healthcare-11-03192-f006:**
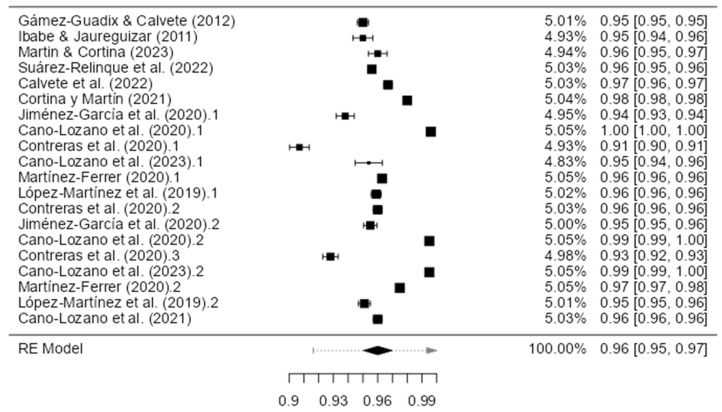
Forest plot of comparative fit index (CFI) (*n* = 20) [15,30,31,33,38,39,42,43,45,46,48,50,52].

**Figure 7 healthcare-11-03192-f007:**
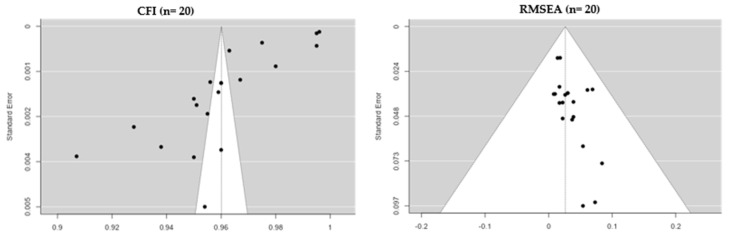
Funnel plot of all studies selected for the meta-analysis of comparative fit index (CFI) and root mean square error of approximation (RMSEA).

**Figure 8 healthcare-11-03192-f008:**
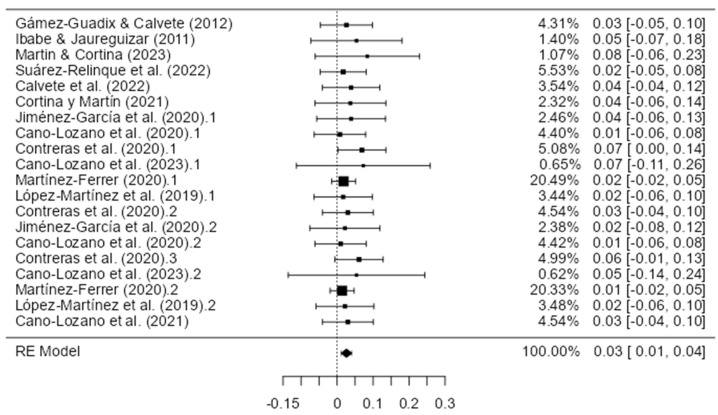
Forest plot of the root mean square error of approximation (RMSEA) (*n* = 20) [15,30,31,33,38,39,42,43,45,46,48,50,52].

**Table 1 healthcare-11-03192-t001:** Characteristics of the articles selected for the study.

N	Author/s	Year	Title	Journal	Q	Instrument	Country	Population	n	Total of Items	Cronbach’s Alpha
1	Bartle-Haring [26]	2015	Reciprocity in Adolescent and Caregiver Violence	Journal of Family Violence	Q2	Conflict Tactics Scale (CTS)	Midwest	Judicial	179	12	G = 0.830
2	Calvete et al. [27]	2011	Child to parent violence in adolescence: environmental and personal characteristics	Infancia y Aprendizaje	Q3	Conflict Tactics Scale (CTS)	Spain	Community	1427	6	G = 0.720
3	Calvete et al. [28]	2015	Predictors of child-to-parent aggression: A 3-year longitudinal study	Developmental Psychology	Q1	Child-to-Parent Aggression Questionnaire (CPA)	Spain	Community	1100	10	T1 G = 0.830 T2 G = 0.840 T3 G = 0.840
4	Calvete et al. [29]	2013	Child-to-Parent Violence: Emotional and Behavioral Predictors	Journal of Interpersonal Violence	Q1	Conflict Tactics Scale (CTS)	Spain	Community	1371	6	T1 G = 0.610 T2 G = 0.680
5	Calvete et al. [15]	2022	The Revised Child-to-Parent Aggressions Questionnaire: An Examination During the COVID-19 Pandemic	Journal of Interpersonal Violence	Q1	Child-to-Parent Aggressions Questionnaire-R (CPA-R)	Spain	Community	1244	10	G = 0.930
6	Cortina and Martín [30]	2021	Validation of the Explanations of Adolescent-to-Parent Violence Scale	Psicothema	Q1	The Explanations about Adolescent-to-Parent Violence Scale (EEVFP)	Spain	Community	763	29	G = 0.920
7	Fernández-Gonzáles et al. [7]	2021	Child-to-Parent Aggression and Dating Violence: Longitudinal Associations and the Predictive Role of Early Maladaptive Schemas	Journal of Family Violence	Q1	Child-to-Parent Aggression Questionnaire (CPA)	Spain	Community	T1 = 1499 T2 = 1262 T3 = 1056	10	T1 G = 0.870 T2 G = 0.840 T3 G = 0.860
8	Gámez-Guadix and Calvete [31]	2012	Child-to-parent violence and its association with exposure to marital violence and parent-to-child violence	Psicothema	Q1	Conflict Tactics Scale (CTS)	Spain	Community	1681	6	G = 0.740
9	Ghanizadeh and Jafari [32]	2010	Risk factors of abuse of parents by their ADHD children	European Child and Adolescent Psychiatry	Q1	Abused Parent Questionnaire	Iran	Clinic	74	27	G = 0.930
10	Ibabe and Jaureguizar [33]	2011	To what extent is child-to-parent violence bi-directional?	Annals of Psychology	Q2	Intra-Family Violence Subscale	Spain	Community	485	9	G = 0.800
11	Ibabe et al. [34]	2014	Behavioral problems and depressive symptomatology as predictors of child-to-parent violence	The European Journal of Psychology	Q1	Intra-Family Violence Subscale	Spain	Clinic and Community	231	6	G = 0.800
12	Ibabe [35]	2014	Direct and indirect effects of family violence on child-to-parent violence / Efectos directos e indirectos de la violencia familiar sobre la violencia filio-parental	Studies in Psychology	Q2	Intra-Family Violence	Spain	Community	485	10	G = 0.800
13	Ibabe [36]	2015	Family predictors of child-to-parent violence: the role of family discipline	Annals of Psychology	Q3	Conflict Tactics Scale (CTS)	Spain	Community	585	16	G = 0.830
14	Martín et al. [37]	2022	Psychosocial Profile of Juvenile and Adult Offenders Who Acknowledge Having Committed Child-to-Parent Violence	International Journal of Environmental Research and Public Health	Q2	Self-Reported Child-to-Parent Violence (APV)	Spain	Judicial	89	9	G = 0.850
15	Martin and Cortina [38]	2023	Profiles of Adolescents who Abuse their Parents: A Gender-based Analysis	Anuario de Psicología Jurídica	Q2	Self-Reported Child-to-Parent Violence (APV)	Spain	Community	341	9	G = 0.690
16	Suárez-Relinque et al. [39]	2022	Emotional Loneliness, Suicidal Ideation, and Alexithymia in Adolescents Who Commit Child-to-Parent Violence	Journal of Interpersonal Violence	Q1	Conflict Tactics Scale (CTS)	Spain	Community	1928	6	G = 0.850
17	López-Martínez et al. [40]	2021	Child-to-Parent Violence, Peer Victimization and Cybervictimization in Spanish Adolescents	International Journal of Environmental Research and Public Health	Q2	Child-to-Parent Aggressions Questionnaire (CPA)	Spain	Community	1318	10	G = 0.850
18	Zuñeda et al. [41]	2016	Characteristics of individuals and families of adolescents in child-to-parent violence: physical aggressiveness, cohesion within the family and interparental conflict as explanatory variables	Revista de Psicopatología y Psicología Clínica	Q3	Conflict Tactics Scale (CTS)	Spain	Community	34	6	G = 0.810
19	Cano-Lozano et al. [42]	2020	Analyzing the Relationship Between Child-to-Parent Violence and Perceived Parental Warmth	Frontiers in Psychol	Q1	Child-to-Parent Violence Questionnaire—Adolescents (CPV)	Spain	Community	1599	14	F = 0.820 M = 0.820
20	Cano-Lozano et al. [43]	2021	Child-to-Parent Violence: Examining the Frequency and Reasons in Spanish Youth	Family Relations	Q1	Child-to-Parent Violence Questionnaire—Youth (CPV)	Spain	Community	1543	19	F = 0.860 M = 0.860
21	Cano-Lozano et al. [44]	2022	Relationship between Punitive Discipline and Child-to-Parent Violence: The Moderating Role of the Context and Implementation of Parenting Practices	International Journal of Environmental Research and Public Health	Q2	Child-to-Parent Violence Questionnaire—Youth (CPV)	Spain	Community	1543	19	F = 0.850 M = 0.850
22	Cano-Lozano et al. [45]	2023	Child-to-parent Violence Offenders (Specialists vs. Generalists): The Role of Direct Victimization at Home	The European Journal of Psychology Applied to Legal Context	Q1	Child-to-Parent Violence Questionnaire—Adolescents (CPV)	Spain	Judicial	208	14	F = 0.900 M = 0.910
23	Contreras et al. [46]	2020	Socio-cognitive variables involved in the relationship between T violence exposure at home and child-to-parent violence	Journal of Adolescence	Q1	Child-to-Parent Violence Questionnaire—Adolescents (CPV)	Spain	Community	1624	14	F = 0.788 M = 0.792
24	Del Hoyo-Bilbao [47]	2020	Multivariate models of child-to-mother violence and child-to-father violence among adolescents	European Journal of Psychology Applied to Legal Context	Q1	Child-to-Parent Aggression Questionnaire (CPA)	Spain	Judicial	288	10	F = 0.870 M = 0.870
25	Jiménez-García et al. [48]	2020	Adaptation and Psychometric Properties of the Child-to-Parent Violence (CPV-Q) in Young Chileans	Revista Iberoamericana de Diagnóstico y Evaluacion Psicologica	Q3	Child-to-Parent Violence Questionnaire—Youth (CPV)	Chile	Community	823	19	F = 0.800 M = 0.800
26	Kuay et al. [49]	2021	Callous-Unemotional Traits are Associated With Child-to-Parent Aggression	International Journal of Offender Therapy and Comparative Criminology	Q1	Conflict Tactics Scale—Father (CTS)	UK and Canada	Community	T1 = 60 T2 = 42	62	T1 F = 0.970 M = 0.890 T2 F = 0.880 M = 0.670
27	López-Martínez et al. [50]	2019	The Role of Parental Communication and Emotional Intelligence in Child-to-Parent Violence	Behavioral Sciences	Q2	Child-to-Parent Aggression Questionnaire (CPA)	Spain	Community	1200	10	F = 0.875 M = 0.875
28	Lyons et al. [51]	2015	Child-to-Parent Violence: Frequency and Family Correlates	Journal of Family Violence	Q1	Conflict Tactics Scale (CTS)	Canada	Community	365	6	F = 0.650 M = 0.640
29	Martínez-Ferrer et al. [52]	2020	Suicidal Ideation, Psychological Distress and Child-To-Parent Violence: A Gender Analysis	Frontiers in Psychology	Q2	Conflict Tactics Scale (CTS)	Mexico	Community	8115	6	F = 0.710 M = 0.710
30	Navas-Martínez and Cano-Lozano [53]	2022	Differential Profile of Specialist Aggressor versus Generalist Aggressor in Child-to-Parent Violence	International Journal of Environmental Research and Public Health	Q2	Child-to-Parent Violence Questionnaire—Adolescents (CPV)	Spain	Community	3142	14	F = 0.660 M = 0.670
31	Orue et al. [54]	2021	Early Maladaptive Schemas and Social Information Processing in Child-to-Parent Aggression	Journal of Interpersonal Violence	Q1	Child-to-Parents Aggression Questionnaire-Father (CPA)	Spain	Community	1256	10	T1 F = 0.730 M = 0.760 T2 F = 0.720 M = 0.750 T3 F = 0.710 M = 0.750

Note. G = global; F = father; M = mother; T = time.

**Table 2 healthcare-11-03192-t002:** Random-effects model and heterogeneity statistics, which reported Cronbach’s alpha globally (*n* = 17), for fathers (*n* = 12), and for mothers (*n* = 12).

	Cronbach’s Alpha		
	Global	Fathers	Mothers
Estimate	0.833	0.800	0.810
SE	0.0129	0.0203	0.0179
Z	64.7	39.6	45.5
*p*	<0.001	<0.001	<0.001
CI Lower Bound	0.808	0.763	0.777
CI Upper Bound	0.858	0.843	0.847
Tau	0.052	0.070	0.061
Tau^2^	0.0027 (SE = 0.001)	0.0049 (SE = 0.0021)	0.0038 (SE = 0.0016)
I^2^	98.45%	98.92%	98.66%
H^2^	72.480	92.167	74.787
df	16	11	11
Q	1104.438	990.231	936.795
p	<0.001	<0.001	<0.001

**Table 3 healthcare-11-03192-t003:** Descriptive analysis of the fit indices of the reviewed studies (*n* = 21).

	CFI	TLI	RMSEA	SRMR
Reported	21	7	21	8
Did not report	0	14	0	13
Mean	0.959	0.821	0.037	0.058
Median	0.959	0.950	0.030	0.061
Standard deviation	0.021	0.344	0.022	0.012
Minimum	0.907	0.041	0.008	0.032
Maximum	0.996	0.980	0.084	0.070
25 percentile	0.950	0.933	0.018	0.057
50 percentile	0.959	0.950	0.030	0.061
75 percentile	0.967	0.955	0.054	0.064

Note. Comparative fit index = CFI, Tucker–Lewis index = TLI, root mean square error of approximation = RMSEA, and standardized root mean square residual = SRMR.

## Data Availability

The data supporting this research can be requested by e-mailing the corresponding author.

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
