# Peer review of "Instruments of Child-to-Parent Violence: Systematic Review and Meta-Analysis"

_healthcare, 2023, doi:10.3390/healthcare11243192_

Round 1
Reviewer 1 Report
Comments and Suggestions for Authors
This manuscript is a meta-analysis to evaluate the internal consistency of instruments for assessing CPV. Although it is an undoubtedly interesting study, it contains various deficiencies and gaps that should be addressed.
Introduction:
In general, the introduction is correct and well-structured. It would be interesting to emphasize the importance of generalization of reliability studies since estimating the reliability of various applications of an instrument provides important information to researchers when deciding whether to use a particular test.
In the introduction, it is stated, "This leads us to propose carrying out an exhaustive analysis of the psychometric properties (internal consistency, content validity, construction validity, capacity to respond to change) of the instruments that are being used to evaluate CPV..." but in the results, only internal consistency has been examined through Cronbach's alpha. This statement should be corrected.
Method:
In the "Search strategies" subsection, it should be mentioned in which fields the keyword searches were conducted (full text, title, and abstract, etc.).
In the "Data extraction process" subsection, it would be helpful to clarify which data (variables) were exactly extracted from the studies included in the meta-analysis. It would also be convenient to comment on whether this process was carried out by a single person or by more than one, and in case more than one person was involved, how discrepancies were resolved.
The eligibility criteria seem very limited. Were publication dates or languages considered? Additionally, since it is not mentioned, I understand that besides peer-reviewed articles, other types of documents (conference proceedings, book chapters, theses, etc.) were also accepted, but this should be clarified.
In the "Statistic analysis" subsection, the software used for performing the meta-analytic analyses is not mentioned.
Results:
While an I2 of 98.33% is indeed high, it would be advisable to clarify the criterion being followed to interpret it.
The forest plot in Figure 2 is not very clear as the rows are very close together, so more space should be left between them. What seems more significant is that the dashed line, indicating the estimated alpha in the meta-analysis, is displaced to the left in all three forest plots.
The fact that the funnel plot exhibits clear asymmetry and, most importantly, the significant value of the Egger regression test, indicates the risk of publication bias. Therefore, it is necessary to consider the result obtained through the trim and fill technique.
I also believe it would be appropriate to conduct a leave-one-out sensitivity analysis to determine if any study is unduly influencing the estimated combined reliability.
I believe the utility of this work would be significantly enhanced if subgroup analyses of the reliability of those tests with a sufficient number of studies to conduct this type of analysis were performed. Thus, we could not only assess whether the instruments for assessing CPV are reliable in general but also which specific instrument exhibit greater reliability.
Discussion
In the discussion (line 214), it is mentioned that the studies span from 2002 to 2018, while the results section mentions studies from 2002 to 2023. This information should be clarified.
In this section, it is stated that moderation analyses of the variables age and gender were conducted. The results of these analyses should be included in the results section, even if they were not significant.
Author Response
Dear reviewer:
We wish to extend our deepest appreciation for the time and effort you dedicated o reviewing our manuscript. Your insights have been closely examined and greatly valued, allowing us to address any pressing concerns toward improving the quality of our work.
Please find attached a response sheet detailing the issues you thoughtfully raised, along with descriptions of modifications made to the manuscript where applicable.
Your guidance throughout this process has proven truly indispensable. We sincerely hope the revisions put forward meet your expectations for standards of quality and scholarly merit. Additional recommendations are still welcomed and will be met with the utmost consideration.
With gratitude,

Reviewer 2 Report
Comments and Suggestions for Authors
The inclusion of "Scoping Study" in the title is somewhat unusual for a study that primarily focuses on systematic review and meta-analysis. It may lead to confusion, as the main research methodology appears to be systematic review and meta-analysis. The term "Meta-analysis Generalization" might not be immediately clear to all readers. A more straightforward and widely recognized term like "Meta-analysis" would provide greater clarity.
The study had a good methodology for conducting a systematic review and meta-analysis. However, it did not fully capitalize on the potential of the gathered data and had a limited focus on Cronbach's alpha as the sole measure of reliability. It's important to assess the broader characteristics of the instruments being analyzed, as relying solely on Cronbach's alpha might not provide a comprehensive picture of their effectiveness.
Further , a clear and detailed explanation of the meta-analysis methods is crucial to ensure transparency, reproducibility, and the validity of the study's findings.
The study could benefit from revisiting its data and reporting more comprehensively on various characteristics and measures used to assess the reliability and validity of the child-to-parent violence instruments. This would help researchers and practitioners gain a more nuanced understanding of these instruments and their applicability in different contexts.
Thank you
Comments on the Quality of English Language.
Author Response
Dear reviewer:
We wish to extend our deepest appreciation for the time and effort you dedicated o reviewing our manuscript. Your insights have been closely examined and greatly valued, allowing us to address any pressing concerns toward improving the quality of our work.
Your guidance throughout this process has proven truly indispensable. We sincerely hope the revisions put forward meet your expectations for standards of quality and scholarly merit. Additional recommendations are still welcomed and will be met with the utmost consideration.
With gratitude,
- The terms "scoping study" will be removed. Furthermore, use of the word "generalization" will also be eliminated. Our intention was to address the meta-analysis results’ capacity for generalization based on the analyzed studies’ I2 values. However, following extensive discussions between authors, we concur with the reviewer’s recommendations and will strike both terms from the title accordingly.
- While we can only report Cronbach's alpha given its predominant use across studies enabling comparisons, we appreciate your recommendation to incorporate additional model fit indices where possible, including CFI, TLI, and RMSEA. Regrettably, such analysis will not encompass all studies. However, the difficulty in extracting further metrics will be noted thoroughly in the limitations alongside the restricted generalizability and depth resulting from sole use of Cronbach's alpha.
- The methodology behind conducting the meta-analysis will be elucidated in detail.
- The English language usage throughout the writing will undergo comprehensive review, with ample improvements provided.
Reviewer 3 Report
Comments and Suggestions for Authors
I sincerely appreciate the opportunity to review your work. I hope that the comments and suggestions I have provided will be helpful in improving your research.
INTRODUCTION
The introduction delves into the complex landscape of Child-to-Parent Violence (CPV), underlining its global prevalence despite being the least understood among various family violence types. Notably, it traces the historical roots of CPV dating back over sixty years, citing Harbin and Madden's early conceptualization as the "battered parent syndrome." Despite this history, CPV remains less recognized compared to other familial violence forms, prompting discussions on terminology, definitions, and evaluation methodologies. It highlights pivotal aspects such as the variability in defining CPV, citing Cottrell and Pereira et al.'s definitions, and points out the challenges in evaluating CPV, referencing Ibabe's identification of assessment instruments. However, it lacks a precise articulation of the specific research objectives and methodologies adopted, a critical component for clarity and focus in the subsequent sections. The section, while comprehensive in laying the groundwork, would benefit from clearer delineation of research objectives and methodologies for enhanced comprehension and focus.
The study addresses Child to Parent Violence (CPV), examining definitions, evaluations and properties of instruments, closing gaps in the understanding of this family violence, with emphasis on terminological variability, to provide essential information to professionals and researchers.
The topic is both relevant and addresses a specific gap in the field. Child-to-Parent Violence (CPV) remains relatively understudied compared to other forms of family violence. The research focuses on evaluating and understanding CPV, including its definitions and assessment methodologies, which is crucial due to the limited understanding and recognition of this form of violence. By delving into the complexities of CPV and analyzing assessment instruments, the study fills a critical gap in the field, providing valuable insights for professionals and researchers working to comprehend and address CPV more effectively.
This research contributes significantly by consolidating diverse aspects of Child-to-Parent Violence (CPV), including its definitions, evaluation tools, and psychometric properties of assessment instruments. Unlike existing materials, it comprehensively synthesizes historical context, terminology variability, and assessment challenges in CPV. Additionally, it offers a systematic review and meta-analysis, providing a consolidated and rigorous analysis of available assessment instruments, which is unique compared to fragmented studies available in the field. This consolidation enhances understanding and aids in selecting appropriate tools for professionals and researchers dealing with CPV.
Method
The methodology has strengths, but could benefit from improvements. Ensure accurate inclusion criteria to maintain consistency; specifying the type of quantitative studies and psychometric instruments desired could refine the search and analysis. Additionally, transparency in the data extraction process using Rayyan should be ensured by having multiple reviewers cross-validate the extracted data. Finally, expand the statistical analysis by exploring possible moderating variables that could affect the reliability of the instrument, such as cultural contexts or sample demographics, providing a deeper understanding of the applicability of the CPV assessment tool. You can add everything mentioned as a limitation if you do not take it into account.
Conclusion
The conclusions align with the evidence presented, emphasizing the reliability of CPV assessment instruments while advocating for a contextualized approach in their selection. However, the conclusion falls short in directly addressing the main question posed in the study. While it acknowledges the importance of further research in understanding CPV and intervening effectively in cases of home abuse, it primarily focuses on the instruments' reliability and their utility in identification and intervention. The suggestions for future research, including exploring additional psychometric properties and evaluating preventive and therapeutic interventions, somewhat connect back to the main question but could explicitly tie into the broader aim of comprehensively understanding and addressing CPV. Therefore, while the conclusions touch on essential aspects, a more direct linkage to the initial research question would enhance their relevance.
References
Yes, they are appropriate

Author Response
Dear reviewer:
We wish to extend our deepest appreciation for the time and effort you dedicated o reviewing our manuscript. Your insights have been closely examined and greatly valued, allowing us to address any pressing concerns toward improving the quality of our work.
Your guidance throughout this process has proven truly indispensable. We sincerely hope the revisions put forward meet your expectations for standards of quality and scholarly merit. Additional recommendations are still welcomed and will be met with the utmost consideration.
With gratitude,
INTRODUCTION
- The objectives and methodology shall be articulated with utmost precision to confer absolute clarity regarding the essence of the study.
METHOD
- The inclusion and exclusion criteria shall be thoroughly explained, in addition to elaborating upon the search methodology and analytical procedures implemented.
- The data extraction process will be expounded upon at considerable length, specifically noting the independent review undertaken by two of the authors.
- Comprehensive descriptions of the statistical analyses utilized shall be provided.
- Incorporating moderating variable analysis extends beyond the defined scope of this article; however, avenues for future research shall be highlighted per reviewer recommendations on potentially worthwhile areas such as cultural contexts.
CONCLUSIONS
- Conclusions will be broadened to align with and critically address the principal objective. Linkages between future directions and said core objective will also be strengthened to bolster relevance and applicability of the study’s concluding remarks
Round 2
Reviewer 1 Report
Comments and Suggestions for Authors
In general, all suggestions have been appropriately addressed.
However, there are two points that I believe remain outstanding. Regarding the software used, it should be specified which specific statistical packages have been used in the R environment. Additionally, a sensitivity analysis has not been conducted to ensure that the results obtained are robust and that there is no excessive influence from any individual study on the estimated reliability.
Author Response
Esteemed reviewer,
We sincerely appreciate your valuable feedback for improving our work.
We specified the name of the R package used.
Regarding the sensitivity analysis, we reported the limitations thereof in this study. We have also added a table to the Appendix.
Note that a sensitivity analysis was not conducted in this study for two reasons. In the research area of child-to-parent violence, specifically in our research group, we have not yet developed an algorithm for decision-making regarding study inclusion and exclusion. By employing the trim-and-fill technique to assess the study influence, we realized the necessity of such an algorithm to avoid reliance on criteria lacking sufficient scientific backing. We commit to developing a sensitivity analysis by groups stratified using an evidence-based algorithm for study selection decisions in a follow-up study.
We deeply appreciate your feedback,
Reviewer 2 Report
Comments and Suggestions for Authors
Authors have addressed all prior comments, significantly strengthening the manuscript. No further revisions are necessary, as the work now stands as a significant contribution to the field.
Comments on the Quality of English Language
.
Author Response
Esteemed reviewer,
We sincerely appreciate your valuable feedback for improving our work.
Please note that we have revised our English writing and hope to substantially improve it.
We deeply appreciate your feedback
